# Analysis of Drought Characteristic of Sichuan Province, Southwestern China

**Yin Zhang** [1,2,3] **, Jun Xia** [3,4]**, Fang Yang** [1,2]**, Dunxian She** [3]**, Lei Zou** [4]**, Si Hong** [3]**, Qiang Wang** [1,2]**, Fei Yuan** [1,2] **and Lixiang Song** [1,2,*]

1    Pearl River Water Resources Research Institute, Pearl River Water Resources Commission, Guangzhou 510611, China
2    Key Laboratory of Pearl River Estuary Regulation and Protection, Ministry of Water Resources, Guangzhou 510611, China
3    State Key Laboratory of Water Resources and Hydropower Engineering Science, Wuhan University, Wuhan 430072, China
4    Key Laboratory of Water Cycle and Related Land Surface Processes, Institute of Geographic Sciences and Natural Resources Research, Chinese Academy of Sciences, Beijing 100101, China
*    Correspondence: dambreak@126.com

**Abstract:** Drought is a widespread and destructive natural hazard and is projected to occur more frequently and intensely, with more severe impacts in a changing environment. In this study, we used the standardized precipitation index (SPI) at various time scales (i.e., 3, 6, and 12 months) to provide an overall view of drought conditions across Sichuan Province, southwestern China, from 1961 to 2016. Then, the relationship between the SPI and the soil moisture anomalies was analyzed. Furthermore, the causes of SPI drought from the perspective of large-scale atmospheric circulation were assessed in the study area. The results showed that most stations with decreasing trends were located in the eastern part of Sichuan Province, while most stations with increasing trends were located in the northwestern part, indicating that the eastern region presented a drying trend, while the northwestern part exhibited a wetting trend. The specific analysis focused on extreme drought indicated an increasing occurrence the probability of extreme drought events, which could induce a high potential drought risk in the study area. The SPI values had a strong relationship with the soil moisture anomalies, and the linear correlation coefficients decreased as the time scale increased. This result indicated that SPI3 (3-month SPI) could be regarded as a good predictor of soil moisture drought. The cross wavelet analysis revealed that the Southern Oscillation Index (SOI) had statistically significant correlations with the SPIs in Sichuan Province. The results of this study are useful for assessing the change in local drought events, which will help reduce the losses caused by drought disasters in Sichuan Province.

**Keywords:** drought; standardized precipitation index; GLDAS; large-scale atmospheric circulation; Sichuan Province

## 1. Introduction

Drought is a gradual phenomenon involving a prolonged shortage of moisture due to insufficient precipitation for a certain length of time; drought may also be caused by a high evapotranspiration rate and/or unsustainable water use [1,2]. It is one of the most harmful natural disasters with severe environmental, economic, and social impacts [3,4]. The spatial and temporal study of drought variability and its potential effects, both regionally and globally, is important and useful for a better understanding of changes in drought and the potential drought risk in different regions; additionally, this information can improve drought resistance, as well as water resource management [5,6]. To date, numerous studies have been conducted to explore the spatial and temporal variations in drought in different regions of the world [7–10]. For example, Dai [11] found that drought has increased since

1950 in many terrestrial areas of the world; Wu et al. [8] indicated that China experienced an overall wetting tendency from 1924 to 1953 and shifted to a drying phase beginning in 1954. It is generally known that changes in drought exhibit large spatial variability due to the various influencing driving factors in different regions [5,12]. Therefore, it is of great interest and significance to investigate the drought variability in specific regions and clarify the potential reasons for this variability.

Accurately describing drought propagation is still a challenging task due to its multiple influencing factors (e.g., climate, hydrology, and socioeconomic status) and the difficulty in clearly determining the beginning, termination, development, and recovery processes. The drought index has been proven to be an efficient way to quantify and monitor drought at regional and global scales (e.g., Dai [11], Spinoni et al. [13], Nguvava et al. [14]). Many drought indices have been developed, including the probabilistic-based drought index (e.g., standardized precipitation index (SPI) [15], standardized precipitation evapotranspiration index (SPEI) [16], and standardized runoff index (SRI) [17]), the simply physical-based drought index (e.g., the Palmer drought severity index (PDSI) [18]), and the drought index developed using multiple statistical models that can reflect drought propagation in different ways (e.g., multivariate standardized drought index (MSDI) [19]. Among these drought indices, the SPI is one of the most widely used because it only requires precipitation data. It is easy to calculate and can reflect drought changes on multiple timescales [5]. The efficiency of the SPI in capturing major drought events and detecting drought variability has been reported by several studies conducted worldwide [20–22]. Moreover, the SPI has been proven to be a good predictor to represent the state of soil moisture [23].

Investigating the influence of potential driving forces of drought is also vital for a better understanding of drought formation and propagation mechanisms and is helpful for drought prediction. Meteorological drought often refers to a deficit in precipitation, which is largely affected by large-scale atmospheric circulation, such as the El Niño-Southern Oscillation (ENSO) [24] and Pacific Decadal Oscillation (PDO) [6]. Among these large-scale atmospheric circulations, ENSO is a typical atmosphere-ocean phenomenon in the tropical Pacific that has a marked influence on the global climate. Furthermore, ENSO has been proven to largely influence precipitation and drought conditions in large areas of China [25,26]. For example, Sun and Yang [27] found that a La Nina event was one of the major impact factors that led to the occurrence of the South China drought in the winter and spring of 2011 due to its impact on the northwestern Pacific subtropical high and Rossby wave pattern, which reduced the amount of moisture transported to southern China. Therefore, in consideration of the differences in drought changes in different regions, the possible correlation between ENSO and droughts in specific regions should be further explored.

Drought happens frequently in China and has exerted large negative impacts on the country's economic development, food security, ecosystems, and water resource management [28–30]. Sichuan Province, which is located in southwestern China, frequently experiences severe drought events, and drought disasters have occurred almost every year in this region since 1949 [31]. Droughts always cause a variety of losses, especially in food security, since Sichuan is a major agricultural province. For example, the midsummer drought in 1997 affected a crop area of $12 \times 10^3$ km$^2$, the different levels of drinking water deficiency affected 7.4 million large livestock, and the direct economic losses amounted to 6 billion US dollars [31]. Furthermore, Sichuan Province experienced a severe drought event in 2006, which prevented the harvest of $3.11 \times 10^3$ km$^2$ of crops and caused direct economic losses greater than 10 billion yuan [32]. Most previous studies related to dry/wet conditions and drought in Sichuan Province focused on the changes in dry/wet spells using precipitation indices (e.g., Huang et al. [33]) and the spatial variability of drought intensity changes using the Z index (e.g., Qi et al. [34]). However, little concern has been paid to extreme drought, the relationship between drought and soil moisture, and its potential driving forces in this area.

In this study, considering the above review and previous results for Sichuan Province, the aims were as follows: (1) to detect the spatial and temporal variation in the spatial and temporal changes in drought, especially extreme drought in Sichuan Province; (2) to explore the relationship between the SPI and soil moisture anomalies; (3) to reveal how the drought of Sichuan correlates with ENSO.

The structure of this study is as follows: Section 2 presents the study area, data used and methods; Section 3 shows the results and discussions, followed by the conclusion in Section 4.

## 2. Data and Methods

### 2.1. Study Area

Sichuan Province (26°03′–34°19′ N, 97°3′–108°31′ E) (Figure 1), with an area of $48.6 \times 10^4$ km², is located in southwestern China. Its topography declines from west to east and from the surrounding mountains to the central basin. The western part of Sichuan Province belongs to the first step of China's terrain and is dominated by plateaus, including the Northwest Sichuan Plateau (4000–5000 m) and the Southwest Sichuan Plateau (3500–4000 m). The eastern part of Sichuan Province is in the second step of China's terrain where the Sichuan Basin (300–700 m) is located. The annual average temperature is 15–19 °C, and the average temperature is 5–8 °C in January and 25–29 °C in July. The average annual precipitation ranges from 500 mm in the northeastern part to 1200 mm in the southeastern part. The precipitation in the rainy season (from May to October) generally accounts for 70 to 90% of the total annual precipitation.

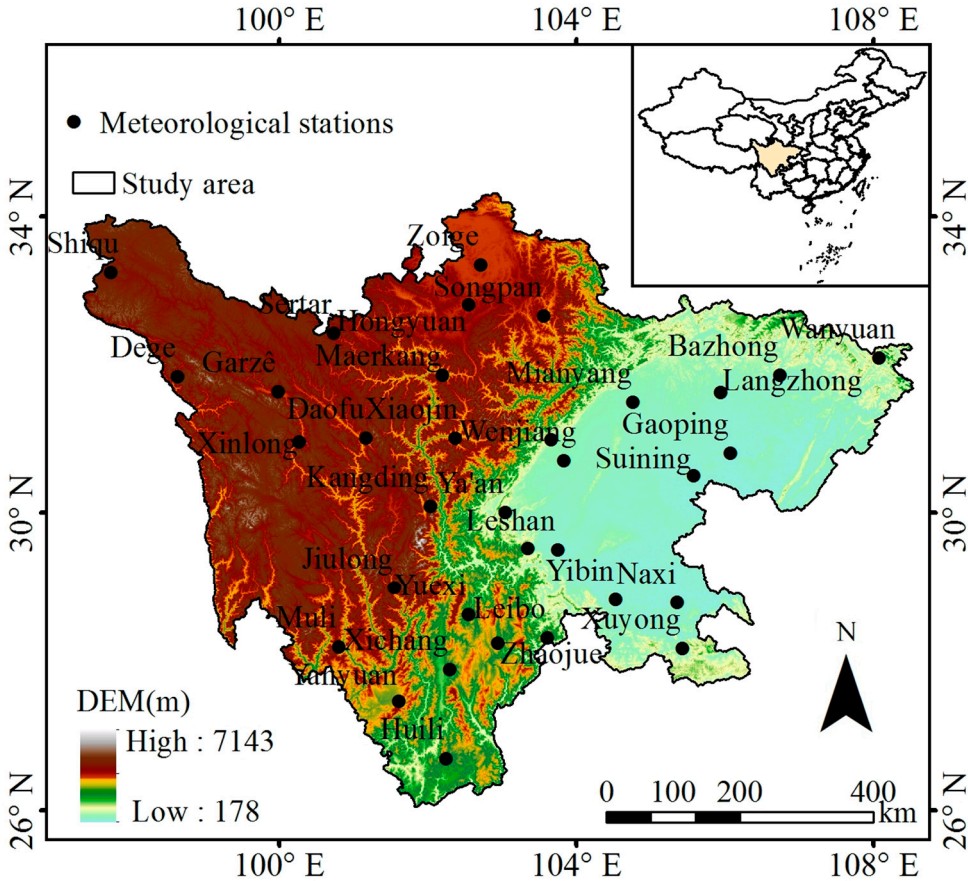

**Figure 1.** Spatial distribution of meteorological stations in Sichuan Province.

*2.2. Data*

The continuous precipitation series from 34 meteorological stations during 1961–2016 were used to analyze the spatiotemporal variation and statistical characteristics of drought in Sichuan Province. These data were provided by the China Meteorological Administration (CMA) and are widely regarded as having high quality. The homogeneity of the datasets was examined before the use of datasets using the RHtests V4 software [35]. The results showed that all the daily precipitation series at 34 meteorological stations passed the homogeneity test at the significance level of 0.05. The regional average precipitation was calculated based on the values at all selected stations in the study area.

Land surface models can provide useful information on the spatiotemporal distribution of soil moisture. The soil moisture field from the Noah land-surface model (Noah-LSM), one of the models available from the Global Land Data Assimilation System (GLDA) dataset (12 h GMT), was adopted due to the long time series (from 1948 to 2010) and high climatological consistency. We used monthly values of GLDAS-2.0 soil moisture from January 1961 to December 2010, with a grid spacing of $0.25° \times 0.25°$. GLDAS-2.0 simulations are solely forced with the Global Meteorological Forcing Dataset from Princeton University. A previous study used GLDAS to investigate the interannual variations in soil moisture in China and found similar results when using in situ observations from the nationwide agrometeorological network [36]. Therefore, it is appropriate to use the GLDAS-2.0 soil moisture dataset in Sichuan Province.

To examine the relationships between drought and large-scale atmospheric circulation patterns, the Southern Oscillation Index (SOI), which is a measure of sea level pressure pattern change during El Niño and La Niño events [37], was used to investigate the influence of ENSO on drought in Sichuan Province. The SOI data from 1961 to 2016 were downloaded from the National Oceanic and Atmospheric Administration (NOAA) Earth System Research Laboratory.

*2.3. Methods*

2.3.1. Standardized Precipitation Index

The SPI was developed by [15] and is a useful index to measure the levels of wetness and drought on different time scales.

First, a gamma probability density function is constructed to fit the frequency distribution of a long-term precipitation series. The gamma distribution function $g(x)$ and cumulative distribution $G(x)$ are as follows:

$$g(x) = \frac{1}{b^a \Gamma(\alpha)} x^{\alpha-1} e^{-x/b} \quad (x > 0) \tag{1}$$

$$G(x) = \frac{1}{b^a \Gamma(\alpha)} \int_0^x x^{\alpha-1} e^{-x/b} dt \quad (x > 0) \tag{2}$$

where $x$ is precipitation, *a* and *b* are the shape and scale parameters, respectively, and $\Gamma(\alpha)$ is a gamma function.

Since the gamma function is not defined for $x = 0$, the cumulative probability function becomes the following:

$$H(x) = q + (1 - q)G(x) \tag{3}$$

where $q$ is the probability that x equals zero.

Finally, the SPI value is computed by transforming $H$ to the standard normal distribution:

$$\text{SPI} = \varphi^{-1}(H) \tag{4}$$

where $\varphi$ is the standard normal distribution.

The short-time-scale SPI describes droughts that affect plant life and farming, while the long-time-scale SPI influences the availability of water supplies/reserves. Agricultural drought is characterized by a 3–6-month time scale, and hydrological drought is character-

ized by a 12–24-month time scale [38,39]. According to precipitation in Sichuan Province, the wet season is defined from May to October since the precipitation during these months generally accounts for 70 to 90% of the total annual precipitation, while the dry season is from November to April. In this study, to analyze the SPI for each season, in the wet (from May to October) and dry (from November to April) periods of the year and on the annual scale, three SPI time scales were selected, 3 months, 6 months, and 12 months, to represent the changes in drought. Specifically, the 3-month SPI was evaluated in May, August, November, and February (seasonal scale: SPI3-spring, SPI3-summer, SPI3-autumn, and SPI3-winter, respectively), the 6-month SPI was evaluated in October and April (wet and dry season, SPI6-wet and SPI6-dry, respectively), and the 12-month SPI was evaluated in December (annual scale: SPI12). Drought classification is provided in Table 1.

**Table 1.** Climate classification according to SPI values.

| SPI Value | Class | Probability (%) |
| --- | --- | --- |
| SPI $\geq$ 2.00 | Extremely wet | 2.3 |
| $1.5 \leq$ SPI $< 2.00$ | Severely wet | 4.4 |
| $1.0 \leq$ SPI $< 1.50$ | Moderately wet | 9.2 |
| $0.00 \leq$ SPI $< 1.00$ | Mildly wet | 34.1 |
| $-1.00 \leq$ SPI $< 0.00$ | Mildly drought | 34.1 |
| $-1.50 \leq$ SPI $< -1.00$ | Moderately drought | 9.2 |
| $-2.00 \leq$ SPI $< -1.50$ | Severely drought | 4.4 |
| SPI $< -2.00$ | Extremely drought | 2.3 |

### 2.3.2. Trend Analyses Methods

An innovative trend analysis (ITA) method, recently proposed by Şen [40,41], has been used to detect trends in environmental, hydrological, and meteorological variables [42,43]. This method requires no strict assumptions (serial correlation, nonnormality, sample number, etc.) and can easily observe the trends of low, medium, and high amounts of data. In the ITA method, a time series is divided into two equal parts and arranged in ascending order. The first subseries ($x_i$) is located on the X-axis, and the second subseries ($y_i$) is on the Y-axis of the Cartesian coordinate system. If the scatter points are collected around the 1:1 (45°) line, there is no trend in the time series. If the scatter points fall in the upper triangular area of the 1:1 line, the time series has an upward trend. In contrast, if the scatter points are in the lower triangular area of the 1:1 line, the time series has a downward trend. In this study, the 10% significance levels were taken as the thresholds to identify significant trends.

Moreover, the nonparametric Mann-Kendall (MK) test is a simple and robust trend detection method [44,45]. It is recommended by the World Meteorological Organization (WMO) as a standard procedure for examining trends in independent hydrometeorological series. In this study, the MK test was used to examine the temporal trends in the extreme precipitation indices, and a significance level of 0.05 was considered. Moreover, the linear regression method, which is a parametric *t*-test method, was used to detect the linear trend pattern of extreme precipitation series.

### 2.3.3. Moving-Window Method

The moving-window method was used to further explore the trends in the changes in the probability of extreme drought. Instead of using whole time series $\{y_i, y_{i+1}, \ldots, y_N\}$, the subseries of $\{y_i, y_{i+1}, \ldots, y_{i+Y-1}\}$ ($i = 1, \ldots, N - Y + 1$) were obtained to apply the fitting. Y is called the moving window size [46]. The selection of an appropriate moving window size is vital for trend detection, and a window size of 30 years was used based on the recommendation of Kao and Ganguly [47] because it was expected to smooth out the effects of most multidecadal climate oscillators and provide more confidence for low frequency extremes [48]. The normal distribution was used to fit the subseries of all SPIs extracted from each of the 30-year moving windows during 1961–2016. Since the subseries subtracted from the moving-window method had strong autocorrelation, it was not appropriate to

use the MK test and linear regression. Therefore, the ITA method, which is not based on a set of assumptions (serial correlation, nonnormality, and sample number, etc.), was used to examine the trends in the probability of extreme droughts.

### 2.3.4. Cross Wavelet Analysis

Wavelet analysis is a useful method for investigating local variations in time series and has been widely used in hydrometeorological series analysis [49–51]. In this study, to investigate how changes in large-scale atmospheric circulation can affect SPI changes in Sichuan Province, cross wavelet analysis [52] was used. The details of the cross wavelet analysis method are as follows.

The cross wavelet spectrum for two time series $A$ and $B$ is given as follows:

$$W_n^{AB}(S) = W_n^A(S)W_n^{B*}(S) \tag{5}$$

where $W_n^A(S)$ and $W_n^B(S)$ represent the wavelet transforms, and $*$ denotes the complex conjugation. The cross wavelet power is defined as $|W_n^A(S)|$.

The theoretical distribution of the cross wavelet power of two time series with background power spectra $P_k^A$ and $P_k^B$ is shown as [52]:

$$D\left(\frac{|W_n^{AB}(S)|}{\sigma_A \sigma_B} < p\right) = \frac{Z_v(p)}{v}\sqrt{P_k^A P_k^B} \tag{6}$$

where $Z_v(p)$ is the confidence level, which is associated with the probability p for a probability density function defined by the square root of the product of two $\chi^2$ distributions. The 5% significance level is selected, namely $Z_v(95\%) = 3.999$.

## 3. Results and Discussion

### 3.1. Spatiotemporal Variations in the SPI

3.1.1. Spatial and Temporal Trends of the SPI

The spatial distribution patterns of the SPIs are presented in Figure 2. The temporal variation in the regionally averaged SPIs in Sichuan Province from 1961 to 2016 is shown in Figure 3.

Concerning the 3-month SPI, only the middle and northwest part showed a lower frequency, while the frequency in the northeast part was as high as 58.92% (Figure 2a). The stations (62%) located in the western part of Sichuan Province showed a higher prevalence of increasing trends, with three stations increasing significantly in spring. The regional average SPI3 time series in Sichuan Province presented a nonsignificant increasing trend at a rate of 0.105/decade in spring (Figure 3a). In summer, both the centers of the areas with the lowest and highest frequency had moved towards the east compared with that in spring, as the low frequency area appeared at the east part while the high frequency area appeared at the northeast corner (Figure 2b). In addition, 53% of stations in Sichuan Province showed decreasing trends, with three stations located in the eastern part of Sichuan Province decreasing significantly. The regional average SPI3 time series decreased slightly at a rate of −0.036/decade in summer (Figure 3b). In autumn, nearly the whole province showed a high drought frequency, and only the areas at the north and south borders showed a lower frequency (Figure 2c). the SPI3 exhibited a slightly decreasing trend over a large part of Sichuan Province. The stations exhibiting increasing trends were situated in the northwestern part of the province. The regional average SPI3 time series in Sichuan Province presented a nonsignificant declining trend at a rate of −0.170/decade in autumn (Figure 3c). In winter, the area with low drought frequency spread from the middle-north to the southwest corner, while an isolated high frequency area appeared at the center of the province (Figure 2d). The SPI3 of half of the stations decreased, that of 44% of the stations increased, and that of 6% of the stations showed no trend. Stations with increasing trends were mainly located in the northern part of the province, while

stations with decreasing trends were mainly located in the southern part. The regional average SPI3 time series increased slightly, at a rate of 0.001/decade in winter (Figure 3d). In the wet season, the area with a high frequency only scattered at the middle-east and middle-west areas (Figure 2e). The proportions of stations with increasing SPI6 trends (47%) and decreasing SPI6 trends (53%) were very similar. Almost all stations in the eastern part exhibited decreasing trends, with two stations having significant decreases; in contrast, the majority of stations in the northwest exhibited increasing trends, and one of them increased significantly. The regional average SPI6 time series in Sichuan Province presented a nonsignificant declining trend at a rate of −0.100/decade in the wet season (Figure 3e). In the dry season, the area with a high frequency concentrated at the middle and western area (Figure 2f). Stations that were mainly located in the western part exhibited increasing trends (56%), with nine of them increasing significantly, while stations that were mainly located in the eastern part exhibited increasing trends (44%), with three of them decreasing significantly. The regional average SPI6 time series in Sichuan Province increased slightly at a rate of 0.098/decade in the dry season (Figure 3f). Considering the 12-month SPI, the area with a high drought frequency only appeared at the middle-east, middle-west part, and the southern border, while the eastern part showed a medium drought frequency (Figure 2g). More stations had decreasing trends (56%), and most were located in the eastern part of the province. The stations with increasing trends were mainly concentrated in the northwest. The regional average SPI12 time series in Sichuan Province presented a nonsignificant decreasing trend at a rate of −0.083/decade (Figure 3g).

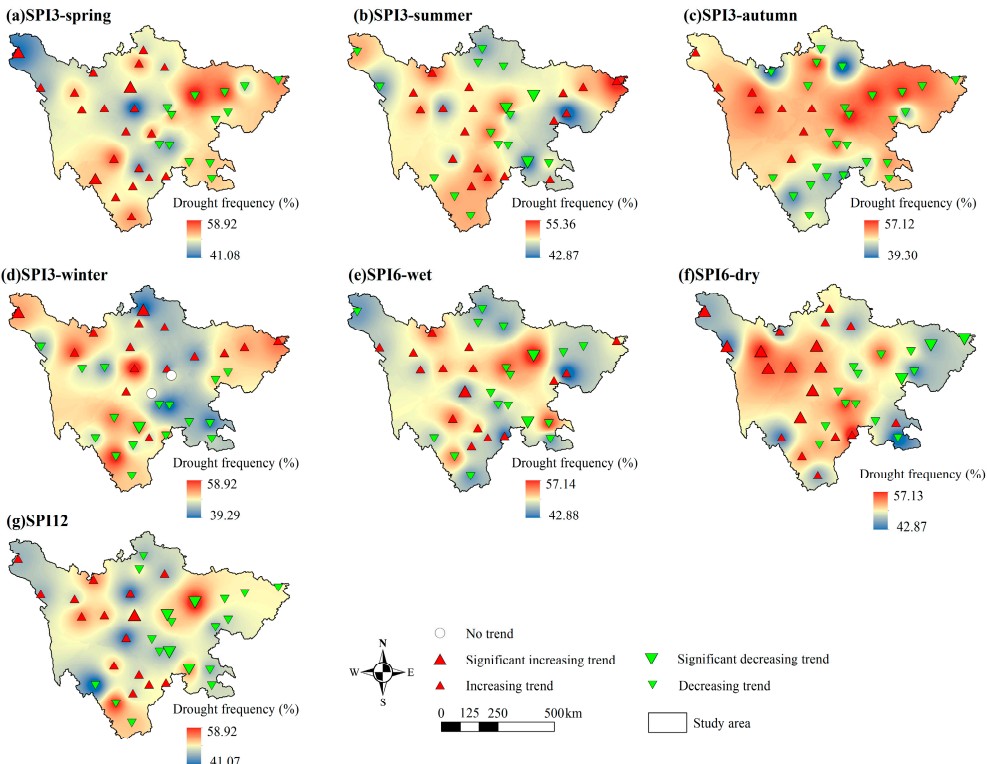

**Figure 2.** Spatial patterns of drought frequency and trends in SPI in Sichuan Province during 1961–2016, (**a**–**d**) for SPI3, (**e**,**f**) for SPI6, and (**g**) for SPI12.

Although the spatial distribution varied with time, the northeast part was more likely to suffer from drought in the SPI3 scenario, as it always showed a higher drought frequency in each season; for the whole province, autumn showed a higher risk of drought. For SPI6 and SPI12 scenarios, the areas at the middle-east and middle-west (the two 'shoulder' areas in the map of Sichuan province) tended to have a higher drought frequency. The trends in SPIs varied spatially, which indicated the heterogeneity in drought variation in Sichuan

Province. Most stations with decreasing trends were clustered in the eastern part of Sichuan Province, while most stations in the northwestern part of Sichuan Province had increasing trends. These results indicate that the eastern part of Sichuan Province was becoming drier, while the northwestern part of Sichuan Province was becoming wetter. Moreover, more attention should be paid to autumn because widespread decreasing trends were detected for SPI3-autumn. Huang et al. [53] used precipitation indices to investigate variations in wet/dry spells in Sichuan Province, revealing the decreasing trend of dry spell indices in western Sichuan Province and the higher risk of drought in autumn, which was basically consistent with our results.

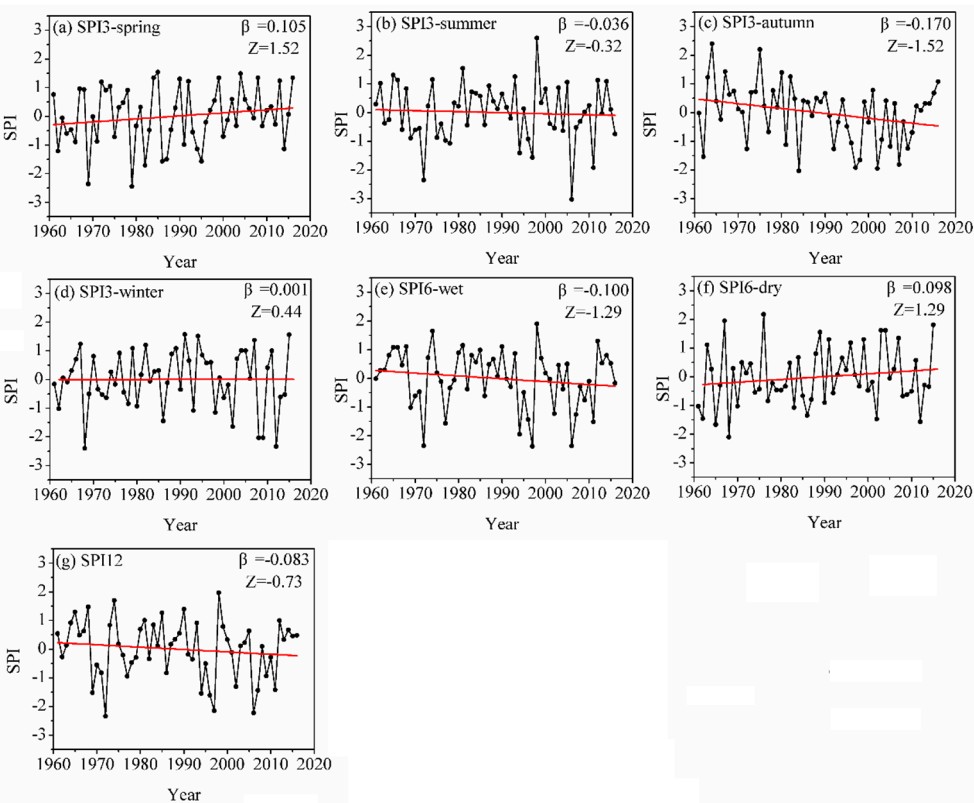

**Figure 3.** Regional average series for SPI in Sichuan Province during 1961–2016. The red line represents the slope of the datasets during 1961–2016 (**a**–**d**) for SPI3, (**e**,**f**) for SPI6, and (**g**) for SPI12.

In Sichuan Province, the magnitude of the precipitation change showed a decreasing trend from the surrounding mountainous regions to the central basin, indicating that the increasing tendency of precipitation was rather prominent at higher elevations, while the decreasing tendency was relatively significant at lower elevations [54–56]. These results may explain why increasing drought trends were mainly found in the eastern part of Sichuan Province, while decreasing drought trends were detected in the northwestern part of Sichuan Province. Moreover, the interannual changes in droughts exhibited obvious seasonality due to precipitation variations. Huang et al. [53] revealed that a decrease in precipitation was found in July, August, September, and October, and a significant increase in drought events was detected in autumn, indicating a higher risk of drought in this season, which was consistent with our results.

The main causes of droughts are atmospheric circulation abnormalities, topographic conditions, and disproportionate precipitation. Regional differences in droughts are related to differences in the geographic position and climate [31]. For example, the western part of Sichuan Province has higher elevations, which result in the average temperatures being 10–12 °C lower and evaporation being 200–400 mm lower. Because of its lower evaporation than precipitation, the western part of Sichuan Province is less likely to experience drought. However, in the eastern part of Sichuan Province, even though there is more precipitation

than that in the western region, the greater evaporation caused by the higher temperatures can accelerate the development of drought, which is why even small precipitation shortages may cause severe droughts in a relatively short time in this area [57]. In terms of the drought distribution in Sichuan, the droughts in the basin and hills are more severe than the droughts in the mountains and plateaus. Therefore, compared with the western region, the east is more vulnerable to drought, and the decrease in precipitation in recent decades further aggravates droughts in the east. The large topographical differences not only affect the climate but also have a certain degree of impact on the population distribution and economic development in different parts of Sichuan Province. The eastern part of Sichuan Province has developed into one of the fastest growing economic zones in western China due to its relatively flat cultivated fields, rich natural resources, and dense population [54]. Since it has a denser population and a more developed economy, frequent droughts will result in more serious consequences in this area. Therefore, more attention should be paid to the eastern part of Sichuan Province because it is more vulnerable to droughts, which could have more severe impacts.

### 3.1.2. Changes in Extreme Droughts

It is well known that extreme drought can have a more intense influence on agricultural systems, water resources, ecological systems, and socioeconomic development than severe or moderate drought events [58–61]; thus, we further investigated the changes in extreme drought in Sichuan Province in this study. Here, extreme drought is defined as a drought event with an SPI $< -2$, corresponding to a cumulative probability function value of 2.3% according to Table 1. We separated the entire time series into two subperiods with the same length, i.e., the first time period was from 1961 to 1988 (denoted as P1) and the second time period was from 1989 to 2016 (denoted as P2). The changes in the occurrence probability of extreme drought events between these two subperiods were examined by using the normal distribution to fit the SPI time series. The efficiency of the normal distribution was tested using the Kolmogorov-Smirnov (K-S) method. This method is based on the difference between the cumulative frequency curve of observation and theoretical frequency curve of expectation [62] and has been widely used [50,63,64]. The results showed that all SPI time series could be well modeled by the normal distribution under the significance level of 0.05.

Figure 4 presents the changes in the probability distribution functions (PDFs) of the two subperiods. The black and red lines denote the annual PDFs in P1 and P2, respectively, and the shaded areas indicate the cumulative probability of extreme droughts during both subperiods. The mean and variance are two important parameters of the normal distribution; specifically, the mean is the location parameter that describes the position of the central tendency, and the variance is the shape parameter that describes the degree of dispersion of the normal distribution. The higher the variance value is, the more dispersed the distribution is. For the PDFs of the SPIs in spring and in the dry season (Figure 4a,f), the mean increased, indicating a wet trend, while the variance decreased, revealing fewer extreme events. For the PDFs of the SPI in summer, the SPI in the wet season, and the 12-month SPI (Figure 4b,e,g), the mean decreased and the variance increased, revealing more extreme drought events. For the PDFs of SPI6-dry (Figure 4f), the mean increased and the variance decreased, revealing fewer extreme events. Comparing the shaded areas of the two subperiods, except in the spring and dry season, most series had a larger shaded area for the latter period, indicating that there were more extreme droughts after 1988 (Table 2).

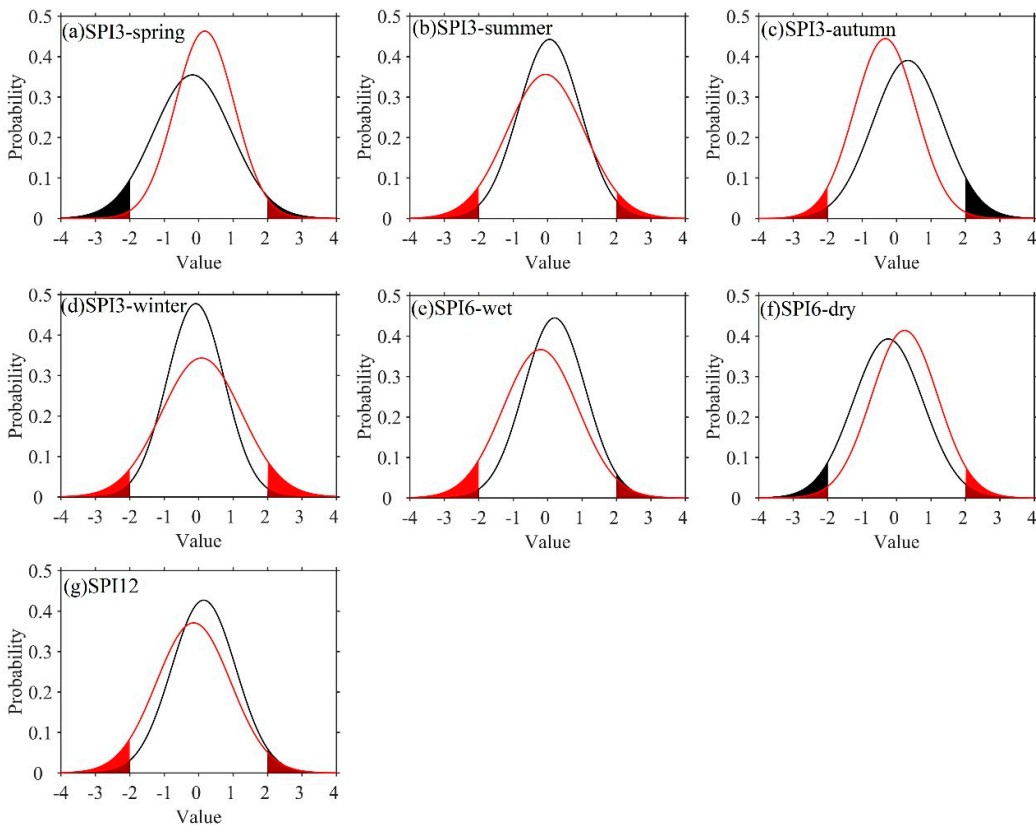

**Figure 4.** Annual PDFs of regionally averaged SPI in 1961–1988 (black line) and 1989–2016 (red line), with shaded areas for extreme drought/wet.

**Table 2.** The probability (%) of extreme drought in 1961–1988 and 1989–2016.

| Period | SPI3-Spring | SPI3-Summer | SPI3-Autumn | SPI3-Winter | SPI6-Wet | SPI6-Dry | SPI12 |
|---|---|---|---|---|---|---|---|
| 1961–1988 | 5.27 | 1.13 | 1.13 | 1.10 | 0.70 | 4.18 | 1.08 |
| 1989–2016 | 0.58 | 4.14 | 3.14 | 3.65 | 4.94 | 1.02 | 4.23 |

Figure 5 presents the trends in the probability of extreme droughts at various time scales using the moving-window method. The probability of extreme droughts in spring and summer exhibited significant decreasing and increasing trends according to the +10% and −10% bands, respectively. In autumn and winter, the probability of extreme droughts showed an increasing trend, and the majority of them increased significantly according to the +10% band. The probability of extreme droughts in the wet season exhibited significant increasing trends according to the +10% band, while the probability of extreme droughts in the dry season exhibited decreasing trends, and most decreased significantly according to the −10% band. For the 12-month SPI, the probability of extreme droughts exhibited an increasing trend, and all increased significantly according to the +10% band. The variations in the probability of extreme droughts varied with different time scales; the extreme droughts increased in the summer, autumn, winter, wet season, and annually, while they decreased in the spring and dry season. In general, extreme droughts became more serious from 1961 to 2016.

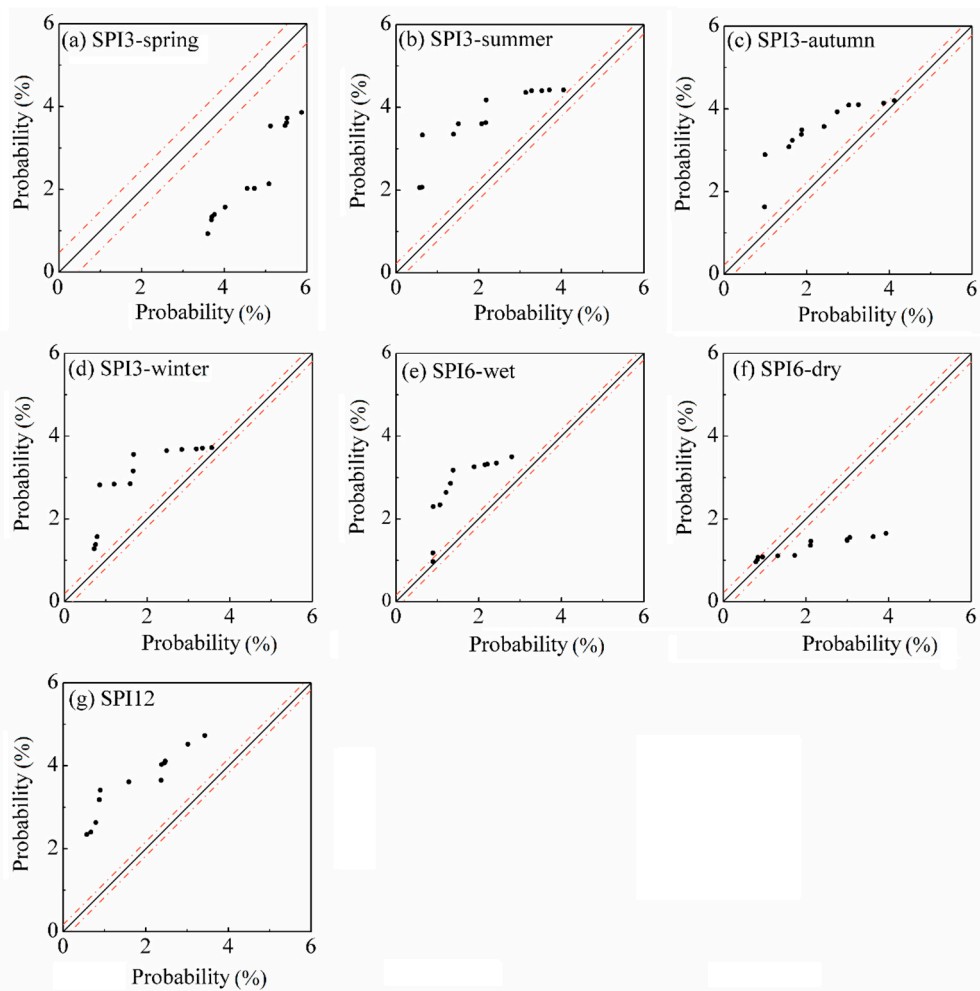

**Figure 5.** The trends of probability of extreme droughts at various time scales using the moving-window method. The x-axis is for probability of extreme droughts from 1962–1991 to 1974–2003 in ascending order; the y-axis is for probability of extreme droughts from 1975–2004 to 1987–2016 in ascending order.

Many previous studies have investigated the extreme drought frequency using the trend analysis method [65–67]; however, we explored the changes in extreme drought from the perspective of changes in the PDF curves. Yang [68] found that extreme severe drought events showed a gradual increasing trend in China, and southwestern China had a more prominent characteristic of clustered extreme droughts than other regions. The increased mean annual frequency of extreme drought was mainly detected in a strip from Southwest China to the western part of Northeast China due to the changes in the sea surface temperature in the tropical Pacific and the tropical Indian Ocean, which resulted in frequent serious drought events in Central and Southwest China [67,69]. Zhang et al. [66] revealed that the interannual and interdecadal trends of extreme drought events exhibited gradually decreasing trends from 1960 to 2013 in the Huaihe River basin, China. In southwestern China, a significant increasing frequency of extreme drought was detected in the southwestern Sichuan Basin from 1960 to 2009 [65]. Moreover, extreme drought increased during the summer monsoon period (May–October), while it decreased during the winter monsoon period (November–April). These studies showed similarities and differences compared with the results of our study, which were due to regional differences.

### 3.2. The Impacts of SPI Drought on Soil Moisture

As a major agricultural province, droughts have large negative impacts on food security in Sichuan [31]. A previous study proved that SPI is a good predictor of crop

production, as it can well represent the soil moisture state [23]. Considering the important role of agriculture in Sichuan Province, we explored the impacts of SPI drought on soil moisture to identify whether it could be used as an alternative tool to monitor agricultural drought in Sichuan Province.

The standardized soil moisture anomaly (SSMA), which eliminates the influence of soil moisture differences between months, was used to characterize the soil moisture anomaly. Figure 6 shows the correlation coefficients between the SPI for the 3-, 6-, and 12-month time scales and the SSMA from GLDAS. At the 0–10 cm layer (surface layer) and the 0–100 cm layer (root-zone layer), the highest correlations (r = 0.59, r = 0.64) were found at the 3-month time scale, while the lowest correlations (r = 0.25, r = 0.51) were found at the 12-month time scale. Moreover, the correlations between the SPIs and SSMA were all significant at the 0.01 level. Generally, SPI3 had the strongest correlation with the soil moisture anomaly, while SPI12 showed the weakest correlation. The strong correlation between SPI3 and SSMA revealed the applicability of these indices to represent agricultural droughts. Some studies have explored the relationship between the SPI and GLDAS soil moisture data in other regions. For example, Spennemann et al. [70] compared GLDAS soil moisture anomalies against the SPI over South America and found that they were strongly correlated.

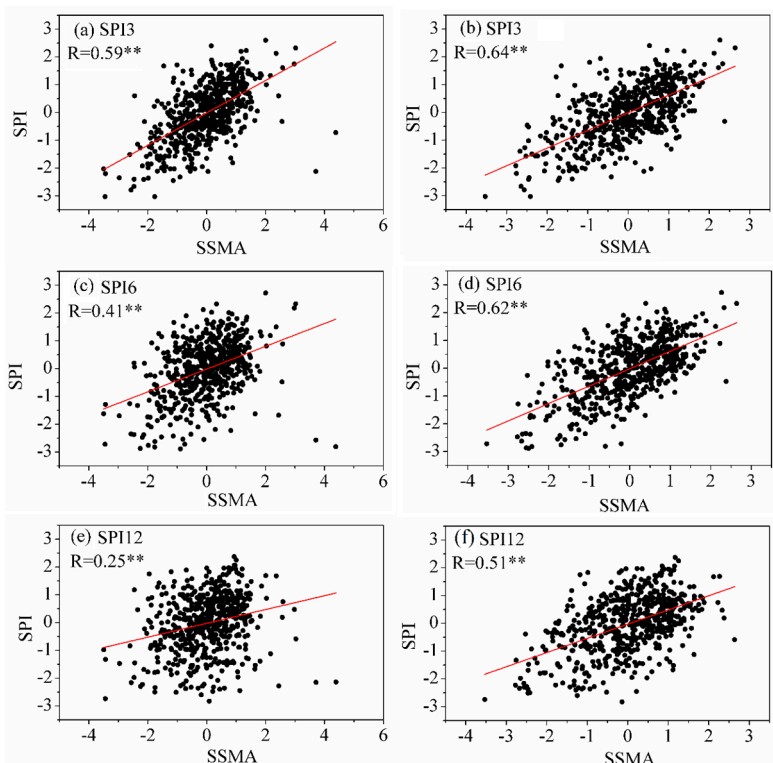

**Figure 6.** Correlation between the SPI at different time scales and SSMA from different layers, (**a**,**c**,**e**) for 0–10 cm and (**b**,**d**,**f**) for 0–100 cm. (** significant at the 0.01 level.).

To further explore the relationship between the SPI and the soil moisture anomaly, extreme events (August 2006) were selected as a case study. In this drought event, the area of crops affected by this drought was $20.7 \times 10^3$ km$^2$, and crops within an area of $3.11 \times 10^3$ km$^2$ were not harvested [32]. In Figure 7a, the SPI3 value of most areas was lower than 0.0, indicating that Sichuan Province showed a general drought in August 2006. Extreme droughts were detected in the southern and eastern regions of Sichuan Province. The 0–100 cm layer soil moisture values for the whole time series (1960–2010) minus August 2006 (left boxplot) and for August 2006 (right boxplot) for GLDAS are shown in Figure 7b. The significant reductions were observed between the boxplots of the complete time series compared to the event for GLDAS (median from 0.31 to 0.28 m$^3$/m$^3$). Similar decreases

were observed in the first quartile (from 0.27 to 0.24 m$^3$/m$^3$) and third quartile (from 0.37 to 0.32 m$^3$/m$^3$) and in the lower whisker (from 0.13 to 0.12 m$^3$/m$^3$) and upper whisker (from 0.43 to 0.40 m$^3$/m$^3$) for GLDAS.

(a)
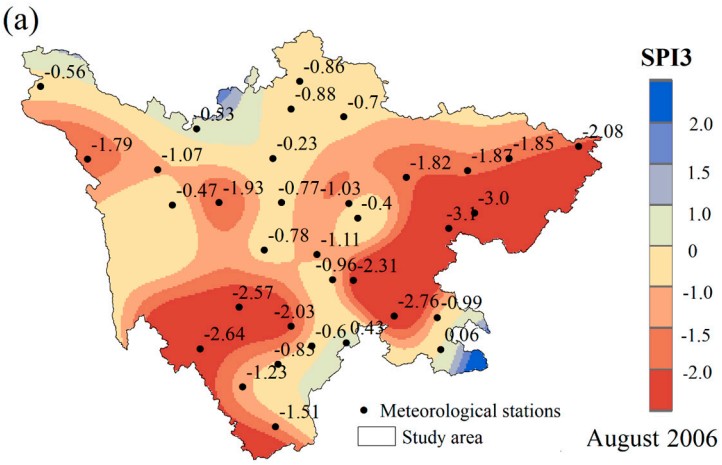

(b)
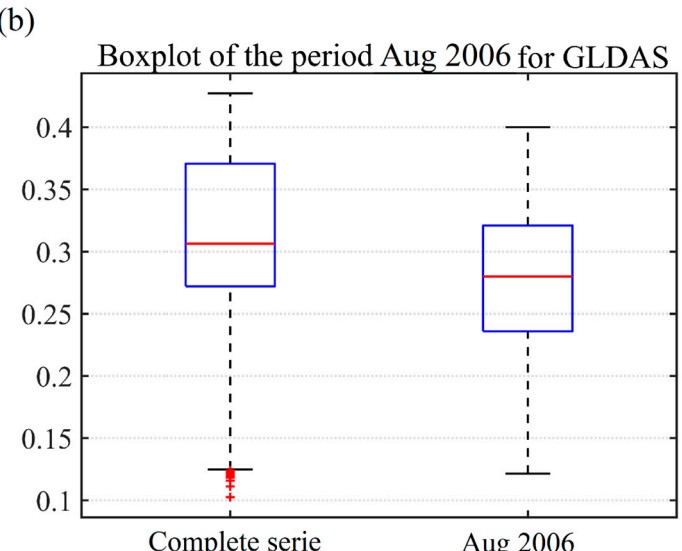

**Figure 7.** August 2006 drought event as seen by SPI and GLDAS (**a**) for SPI in August 2006 and (**b**) for the boxplot of the period of August 2006 for GLDAS (upper and lower whiskers, 95th and 5th percentile value; upper and lower borders of the box, the upper quartile and lower quartile; red line, medium value).

Abnormal climate, sudden decreases in precipitation, abnormally high temperatures, and high evaporation levels were the main causes of this drought. Sichuan Province frequently experiences widespread droughts that lead to serious damage and high losses. In addition to the drought mentioned above, drought disasters in Sichuan Province occur almost every year, causing serious environmental, economic and social losses. Therefore, it is vital to take action to reduce and eliminate the damage and losses caused by droughts by adopting comprehensive prevention and remediation measures of water retraining, cultivation change, and forestation. The SPI is a good tool to monitor agricultural droughts in Sichuan Province because it has a strong correlation with soil moisture and can capture drought events.

### 3.3. Relationship between the SPI and Large-Scale Atmospheric Circulation

In this study, the SOI was adopted to explore the possible impact of ENSO on drought in Sichuan Province. SPI3-summer, SPI6-wet, and SPI12 were selected as representative indices to further reveal the time–frequency relationship between droughts and SOI.

Figure 8 presents the cross-wavelet transform of the SPI and SOI. Figure 8a shows that the SOI had a 5–8-year signal of positive correlation from 1970 to 1980, a 9–13-year signal of positive correlation from 1973 to 1981, and a 2-year oscillation signal from 1996 to 1998. Figure 8b suggests that the SOI had a 2-year oscillation signal from 1970 to 1971, a 5–7-year signal of positive correlation from 1972 to 1980, an 8–13-year signal of positive correlation from 1986 to 2003, and a 2-year oscillation signal from 1996 to 1998. Six statistically significant positive correlations between SOI and SPI12 are shown in Figure 8c: a 2–3 signal from 1966 to 1972, a 6-year signal from 1972 to 1978, a 5-year signal from 1983 to 1987, a 2-year signal from 1994 to 1996, a 6-year signal from 1994 to 2000, and an 11–12-year signal from 1994 to 2001. These results indicated that the SOI had a strong association with changes in droughts in Sichuan Province, which demonstrated that the droughts in Sichuan Province were influenced significantly by the SOI.

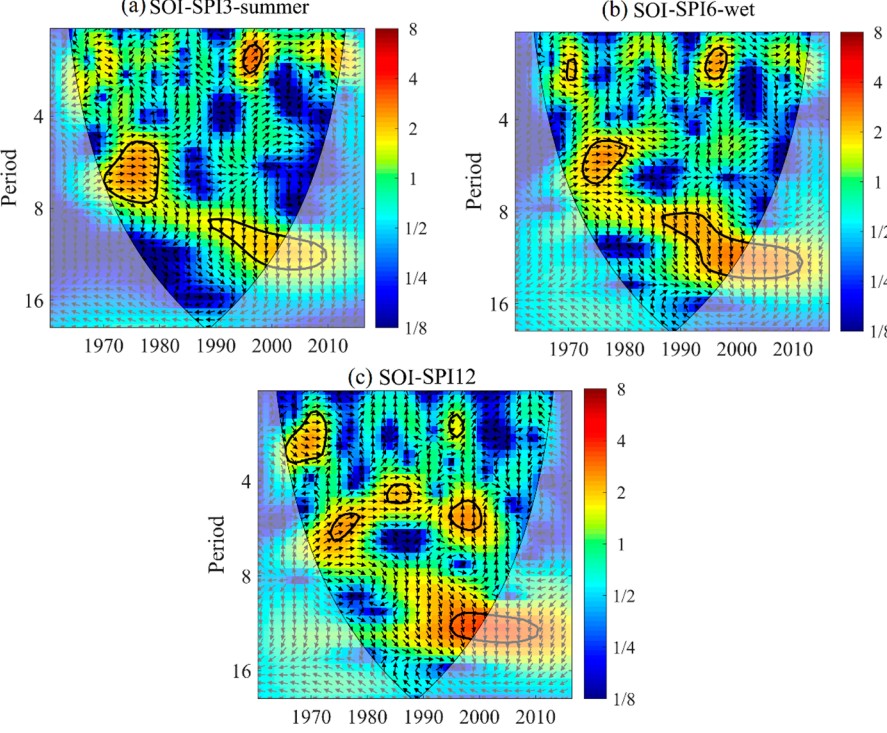

**Figure 8.** Cross-wavelet transforms of SOI and SPI3-summer (**a**), SOI and SPI6-wet (**b**), and SOI and SPI12 (**c**) time series. The thick black contours depict the 5% confidence level, and the black line is the cone of influence. Right-pointing arrows indicate that the two signals are in phase, while left-pointing arrows indicate anti-phase signals.

A previous study proved that ENSO has a marked influence on precipitation in Sichuan Province [56]. Precipitation decreases during the warm-phase periods of El Niño events, while it increases during the cool-phase periods of La Niña events in Sichuan Province [71]. Since deficits in precipitation are a major cause of droughts, ENSO plays an important role in inducing droughts in Sichuan Province. Moreover, ENSO has been proven to have a significant impact on droughts in various regions of China. ENSO has a strong impact on inducing droughts in the Yangtze River basin [72], Yellow River basin [53,73], and Wei River basin [74]. Several studies have focused on the influencing mechanism of ENSO on drought in China. During El Niño events, anomalous anticyclonic flow over the western North Pacific and anomalous southwesterly winds along the southeast coast of China

in the lower troposphere enhance the moisture supply and increase rainfall in southern and eastern China, leading to a decrease in the probability of drought occurrence in these areas [37,75,76]. The anomalous East Asian cyclone is displaced southwestward during the developing phase of ENSO [77], which results in a decrease in summer/autumn rainfall in northern China, further leading to droughts in this area [37]. Although cross-wavelet transforms are used to explore the relationship between SPIs and ENSO, more tools or methods need to be applied for the relevant research for a more comprehensive understanding of the relationship between SPIs and other large-scale atmospheric circulations and the drought formulation mechanism for further research.

## 4. Conclusions

In this study, based on the daily precipitation data of 34 meteorological stations in Sichuan Province from 1961 to 2016, three different SPI time scales were selected to analyze the spatiotemporal variation in drought. Their correlation with soil moisture and large-scale atmospheric circulation were also explored. The main conclusions are presented as follows:

(1) At the spatial scale, most stations with decreasing trends were clustered in the eastern part of Sichuan Province, while most of the stations in the northwest region exhibited increasing trends. These results indicated that the northwest region had a wetting tendency, while the eastern part of Sichuan Province had a drying trend. Comparing the PDFs of regionally averaged SPIs before and after 1988, except in the spring and dry season, most series had a larger shaded area in the latter period, indicating that there were more extreme droughts after 1988. The same results were found based on trends in the probability of extreme droughts using the moving-window method, indicating that Sichuan Province is likely to experience more extreme droughts.

(2) The SPIs had a strong relationship between the SSMAs, especially SPI3, indicating that this index can reveal agricultural droughts to some extent. In August 2006, the spatial variation in SPI3 showed that most areas of Sichuan Province experienced droughts, especially the southwestern and eastern regions. The boxplot of soil moisture showed that soil moisture was lower in August 2006 than in the complete series, indicating that drought occurred in August 2006. These results confirm that the SPI is a proper predictor of soil moisture in Sichuan Province and can reveal agricultural droughts to some extent.

(3) The SOI had a statistically significant correlation with SPI3-summer, SPI6-wet, and SPI12, which demonstrated the short-, medium-, and long-term droughts in Sichuan Province, respectively. That is, the SOI had a strong influence on inducing droughts in Sichuan Province. This result demonstrated that the SOI has the potential to improve the prediction of drought events in the study region.

**Author Contributions:** Conceptualization Y.Z. and L.S.; methodology, Y.Z.; data curation, Y.Z., F.Y. (Fang Yang), and L.Z.; formal analysis, Y.Z. and J.X.; writing—original draft preparation, Y.Z., L.S. and Q.W.; writing—review and editing, Y.Z., D.S. and S.H.; visualization, Y.Z. and F.Y. (Fei Yuan). All authors have read and agreed to the published version of the manuscript.

**Funding:** This research was supported by the National Key R&D Program of China (2021YFC3001000) and the Strategic Priority Research Program of the Chinese Academy of Sciences (XDA23040304).

**Data Availability Statement:** Not applicable.

**Acknowledgments:** The authors thank the China Meteorological Administration, Princeton University, and National Oceanic and Atmospheric Administration (NOAA) Earth System Research Laboratory for access to data.

**Conflicts of Interest:** The authors declare no conflict of interest.

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
