# Peer review of "Analysis of Drought Characteristic of Sichuan Province, Southwestern China"

_water, doi:10.3390/w15081601_

Round 1

Reviewer 1 Report

#Major comments#

1.      There are available long-term reanalysis or observational datasets to calculate SPI. If the recalculated SPI analysis with the datasets is consistent with the station-based SPI, it can evidently address.

2.      The SPEI has a better aspect compared to the SPI, but it is still curious that the SPI really represents the drought condition in the regional scale analysis.

3.      It is necessary to describe how to define SPI-3, -6, and -12 (regarding the appended number) and why these indices are adopted in this study, not for using the other numbered SPI.

4.      SPI is entirely based on precipitation, and the correlation between SPI and rainfall could be overestimated because both variables are not independent. Is such a highly correlated relationship also sustained in the correlation based on the independent proxy (temperature/evapotranspiration)?

5.      The authors selected the SPI as the key index for drought rather than the SPEI. A reader may question why the SPI is good for presenting drought events. Any evaluation of SPI in the study region with observed drought events or scientific advantage of using SPI is a need in the manuscript.

6.      According to the descriptions, we know that the work is mainly based on in-situ observations. How did you obtain the regional frequency, duration, and severity for each target region

7.      In figure 2, the authors showed the spatial distribution of the SPI  trend. Looking at this figure, we can see an upward and downward trend. But in the next figure, they draw regional average time series. I am wondering how they get the regional average SPI trend. If the average trend over the region is calculated, some station-based trend signal disappears. Please use a spatial map instead of regional trend lines or describe how it calculates.

8.      Figure captions are not descriptive; therefore, readers can’t understand figures clearly. Add informative figure captions.

#Minor comments#

9.      L282 -“These results indicate the eastern part of Sichuan Province 282 was becoming dry, and the northwestern part of Sichuan Province was becoming wet” how do you define dry and wet? What is the threshold or criteria for identifying wet or dry?

10.  L328 “We separate the entire time series into two subperiods with 328 the same length, i.e., the first time period was from 1961 to 1988 (denoted as P1) and the 329 second time period was from 1989 to 2016 (denoted as P2)”. What is the basis for separating the two periods? Did you use some statistical tool to identify the change point?

11.  L413 20.7 thousand km2, change it like 20.7×103 km

12.  L424 describes the boxplot accurately (red line- mean or median Whiskers –min and max or 5th and 95 percentile

13.  L220 cross-wavelet spectrum analysis for drought and large-scale oceanic circulation is common. Please add some new references to the methodology section

Shelton, S., Ogou, F.K. & Pushpawela, B. Spatial-Temporal Variability of Droughts during Two Cropping Seasons in Sri Lanka and Its Possible Mechanisms. Asia-Pacific J Atmos Sci 58, 127–144 (2022). https://doi.org/10.1007/s13143-021-00239-0

14.  Discussion and conclusion. It is better to separate this section into a discussion and conclusion sections.

Reviewer 2 Report

Please some minor comments in the attached document for your attention and possible action. Great job done

Reviewer 3 Report

The study examines the status of drought conditions across Sichuan Province, southwestern China, from 1961 to 2016. The study succeeds in interpretation of precipitation distribution and current droughts status and strategies to mitigation. Overall, this research is potentially full of interest, as it addresses the relevant topic of the status of droughts; reduce the losses caused by drought disasters in Sichuan Province. Moreover, there is still a lack of examples of this type of research. I believe that the study is drafted in a clear and intelligible fashion; the text is well organized.

  Below are some comments that the authors may needs to address.

Major Issues
1.Why did the authors choose the data from 1961-2016 and not up to 2022?

 2. Limitations of the study need a mention in the abstract and the manuscript. Limitation needs a separate mention; however it is a part of discussion section.
3. The introduction section, apart from introducing the reader to the problem statement, needs an introduction on how the manuscript is arranged to understand the flow. More work is needed from the authors for the same.
4. The literature review section is not sufficient. Literature has been cited in the introduction section. However, the study's relevance increases significantly as the literature review helps understand how the current studies, even for the
Sichuan province, support sustainable goals and identify the gap for readers, researchers, and decision-makers. The literature review can help identify the contradictory thoughts, perspectives, and theoretical implications (if any) while also scrutinizing the gap in the present research. How does current research comply or contradict the earlier thoughts? The literature in the introduction and discussion sections can then support the LR to build a case for the reader to understand the research problem statement precisely.
- Why and how current practices will be impacted by understanding precipitation trends
- Why understanding the key variables and trends is important to understand the impact on
Sichuan province?
- what is the research worldwide talking about the same?
5. The precipitation tends should be explored season wise. Is there any shift in the precipitation trend of the summer, autumn, winter or spring seasons in the Sichuan province? Which season is more prone to drought? And specify the regions also.
This will help the manuscript to give more value addition to the readers.

Minor Issue
Quality of figure 1 needs a check and should be enlarged for better readability.

Wishing you good luck

Round 2

Reviewer 1 Report

The authors' responses to the raised questions are satisfactory and address all comments. Therefore, I would like to accept this article for publication.

Reviewer 3 Report

The comments are adequately addressed and suitable for publication